# Soil Fertility Improvement and Carbon Sequestration through Exogenous Organic Matter and Biostimulant Application

Bozena Debska [1], Karol Kotwica [2], Magdalena Banach-Szott [1,*], Ewa Spychaj-Fabisiak [1] and Erika Tobiašová [3]

[1] Department of Biogeochemistry and Soil Science, Bydgoszcz University of Science and Technology, Bernardynska 6/8 St., 85-029 Bydgoszcz, Poland
[2] Department of Agronomy, Bydgoszcz University of Science and Technology, Al. Prof. S. Kaliskiego 7, 85-796 Bydgoszcz, Poland
[3] Department of Soil Science, Slovak University of Agriculture in Nitra, Tr. A. Hlinku 2 St., 949 76 Nitra, Slovakia
* Correspondence: mbanach@pbs.edu.pl

**Abstract:** One of the main tasks in the search for environmentally friendly crop-growing methods is to increase soil fertility by improving its physical, chemical and biological parameters. The aim of this study was to determine the effect that the long-term annual application of different types of soil fertility agents (exogenous organic matter: 1. manure, 2. straw in combination with nitrogen fertilization and liming and 3. the addition of biostimulants) had on organic matter properties, including humic acid (HAs) properties. The research was carried out on the basis of soil samples from a ten-year pot experiment which was set up as single-factor pot experiment with four replications. PVC pots with perforated bottoms were filled with soil samples taken from the tilled layer of an arable field where winter wheat was grown in monoculture. The pots were exposed directly to the weather and were left without vegetation. The soil samples were assayed for the content of total organic carbon (TOC), total nitrogen and fractional composition of humus. HAs were extracted with the Schnitzer method and analyzed for the elemental composition, spectrometric parameters in the FT-IR and UV-VIS range and hydrophilic and hydrophobic properties. In addition, EPR spectra were produced. The results showed that the content of organic matter compared to soil without additives increased with the use of manure and the use of straw in the CaO variant and in the form of a mulch. The content of dissolved organic carbon (DOC) ranged from 124.6 to 286.1 mg kg$^{-1}$ and had strong positive correlation with TOC content. The values of the ratio of carbon content in humic acids to carbon content in fulvic acids (CHAs/CFAs) ranged from 0.71 to 0.99. The use of a biostimulator—with or without the addition of straw—increased carbon sequestration in humic acid molecules, as well as their oxidation level and their share of hydrophobic fractions with the longest retention time. Thus, the addition of UGmax intensifies humification processes, leading to the formation of stable humic acid molecules.

**Keywords:** fractional composition; humic acids; manure; straw; UGmax

## 1. Introduction

Soil organic carbon (SOC) and nitrogen (N) are two of the most important indicators for agricultural productivity. Soil SOC and N dynamics are governed by climate change, the soil environment and human activities, mainly field management practices [1]. The C and N cycle is important not only for improving crop efficiency, but also for mitigating climate change and the functioning of ecosystems [2]. From an agricultural point of view, a big challenge is to increase or maintain yields without progressive degradation of the Earth's environmental systems, especially soils [3]. Progressing global soil degradation as a result of agricultural intensification [4,5] is further exacerbated by climate change [6,7]. The European Commission has developed a comprehensive green action plan to achieve the European Green Deal target of 25% of agricultural land being farmed organically by

2030 [8]. The aims of this include combating climate change, ensuring environmental protection and preserving biodiversity. The knowledge about organic farming, despite being plentiful, clearly still needs to be further augmented to make practices even more sustainable and more productive.

One of the main tasks in the search for environmentally friendly crop-growing methods is to increase soil fertility by improving its physical, chemical and biological parameters. One of the main components determining soil fertility is the organic matter (OM) content. Sustainable management of organic matter in agriculture relies on increasing the contribution of MO to the soil while reducing its losses [9].

Organic matter consists of fractions of various stabilities (resistance to decay): labile fractions that include dissolved organic matter (DOM) and fractions more resistant to decay, such as fulvic acids (FAs), humic acids (HAs) and humin (h) [10–17].

The most mobile and rapidly decomposing fraction is DOM—more precisely, water-extractable organic matter (WEOM), whose content is determined based on the carbon content in water extracts, i.e., dissolved organic carbon (DOC). In arable soils, DOC generally comprises less than 1% of TOC [18]. Despite constituting such a small share, DOM plays an important role in, inter alia, the biogeochemical carbon, nitrogen and phosphorus cycle, and can be a source of nutrients for microorganisms [14,18–21]. It is generally assumed that changes in DOC content are an important indicator of changes in soils, especially anthropogenic changes [21].

An equally important parameter used to determine the quality of organic matter is CHAs/CFAs—the ratio of carbon content in humic acids to carbon content in fulvic acids [10]. The CHAs/CFAs ratio has been used as an indicator to describe the humification degree of OM, with a larger value indicating a higher degree of humification. It has also been shown that soils with higher CHAs/CFAs are classified as more fertile [10,12,22,23].

The content and quality of organic matter (including the content of DOC, HAs, FAs and humin fractions) are shaped by habitat conditions (temperature and humidity) and anthropogenic conditions, as well as soil management [12,14]. In arable soils, post-harvest residues and fertilization (organic and mineral) are important determinants of the content and quality of organic matter [10,13,24–29].

Of the organic matter (OM) fractions, a major determinant of physical, chemical and biological soil properties is the fraction of humic substances that are soluble in an alkaline environment and non-soluble in an acidic environment—i.e., humic acids (HAs). According to the literature, the determinants of humic acid properties include elemental composition, UV-VIS and IR spectrometric properties, $^{13}$C NMR and chromatographic properties (HPLC and GCMS) [26,30–36]. The properties of HAs are determined by the plants selected in crop rotations and the use of natural (manure, liquid manure and slurry), organic (compost and green fertilizers) and mineral fertilizers [10,12,26,29,37–41].

In recent years, bio-fertilizers (biostimulants) have become increasingly popular [42–46] and the cited authors state that using bio-fertilizers undoubtedly increases crop yields, mainly by increasing nutrient availabilities.

The influence of bio-fertilizers on the physicochemical properties of soil is not unambiguous [13,44,46–48]. Pranagal et al. [46] report that using bio-fertilizers reduces TOC. Debska et al. [13] obtained an increase in carbon content in soil treated with a bio-fertilizer compared to control; moreover, after the application of the bio-fertilizer, the soil had a higher absolute and proportional content of CHAs and humins, a higher CHAs/CFAs ratio and lower DOC.

The aim of this study was to determine the effect of long-term use of (1) exogenous organic matter (manure and straw), (2) mineral fertilization (CO (NH$_2$)$_2$), (3) liming and (4) a biostimulant (UGmax) on the properties of organic matter including the properties of humic acids. It was assumed that such a wide spectrum of fertilization variants, carried out in one experiment under the same soil and climatic conditions, would also allow to determine the role of individual fertilizers in shaping the properties of soil organic matter.

## 2. Materials and Methods

### 2.1. Description of the Study Materials

The types of soil fertility agents analyzed in the study and the methods of their application, which together constitute nine factors, are summarized in Table 1.

**Table 1.** Resources shaping soil fertility and methods of their application.

| Symbol | Object of Experiment |
|---|---|
| A * | Mulch of chopped residual surface straw from the pot |
| B | Chopped straw + CaO mixed with soil |
| C | Chopped straw + Nmin. ** mixed with soil |
| D | Chopped straw + UGmax *** mixed with soil |
| E | UGmax mixed with soil |
| F | Chopped straw mixed with soil |
| G | Manure mixed with soil |
| H | CaO mixed with soil |
| K | Control |

* For this variant, samples were taken from two depths: A1: 0–10 cm; A2: 10–20 cm. ** Mineral nitrogen $(CO(NH_2)_2)$. *** Biostimulant [13].

The soil samples were taken from the tilled layer of an arable field (0–20 cm) where winter wheat was grown in monoculture. According to the WRB classification [49], the sampled soil was classified as Luvisol, with a granulometric composition characteristic for light clay. The ten-year pot experiment was set up as a single-factor pot experiment with four replications. PVC pots (V = 14.72 $dm^3$; h = 30 cm; r = 12.5 cm) with perforated bottoms were filled with soil samples (15 kg each). The pots were placed in the field in a completely random manner and dug in to a depth of 25 cm. The experiment was located at Kicko (N:52°36′30.1″ and E:18°24′00.2″) in the Kuyavian-Pomeranian Voivodeship, Poland. Soil fertility agents were applied in the following amounts in the first decade of September each year.

| | | |
|---|---|---|
| Chopped Straw | 5.0 t $h^{-1}$ | 25 g $pot^{-1}$ |
| UGmax (0.9 $dm^3$ UGmax to 600 $dm^3$ water) | 600 $dm^3$ $ha^{-1}$ | 0.003 $dm^3 pot^{-1}$ |
| CaO | 1500 kg $ha^{-1}$ | 7.5 g $pot^{-1}$ |
| Nmin. $(CO(NH_2)_2)$ | 30 kg $ha^{-1}$ | 0.15 g $pot^{-1}$ |
| Cattle manure | 30 t $ha^{-1}$ | 150 g $pot^{-1}$ |

Each soil application was performed by mixing the substance (except for mulch) into the soil. Until the next application, no activities were performed in the pots except for removing emerging vegetation using glyphosate and keeping the surface free from vegetation. The pots were exposed directly to the weather. The test samples were collected once after 10 years of the experiment, by the vases' liquidation. Soil samples were dried at room temperature and sieved (2 mm).

### 2.2. Methods

#### 2.2.1. Basic Soil Parameters

For air-dried soil samples, the following analyses were made.

The content of total organic carbon (TOC) and total nitrogen (Nt) expressed in g $kg^{-1}$ of d.w. of soil was analyzed with a Vario Max CN analyzer supplied by Elementar (Germany).

The content of dissolved organic carbon (DOC) and dissolved nitrogen (DNt) were assayed in solutions from an extraction of soil sample using 0.004 mol $dm^{-3}$ $CaCl_2$, at a soil-sample-to-extractant ratio of 1:10; extraction took 1 h. The contents of DOC and DNt were assayed with an Analityk Jena Muli N/C 3100 analyzer and expressed in mg $kg^{-1}$ d.w. of soil sample and as percentage share in the pool of TOC and Nt, respectively.

### 2.2.2. Fractional Composition of Humus and Isolation of Humic Acids

The fractional composition of humus was assayed based on the carbon (nitrogen) fractions determined in the extracts using a Multi N/C 3100 from Analityk Jena (Germany), according to the following procedure.

Decalcification (24 h) with 0.05 M HCl (1:10 *w/v*), Cd, (Nd)—carbon (nitrogen)—in solutions after decalcification.

Extraction (24 h) of the remaining solid with 0.5 M NaOH (1:10 *w/v*) with occasional mixing, followed by centrifugation; C(N)HAs + FAs—sum of the carbon (nitrogen) of humic and fulvic acids.

Precipitation (24 h) of humic acids from the resulting alkaline extract with 2 M HCl to pH = 2 and centrifugation; C(N)FAs—carbon of fulvic acids in solutions.

Purification of the resulting humic acids was as follows: the humic acid residue was treated with a mixture of HCl/HF (990 mL $H_2O$, 5 mL HCl, 5 mL HF) over a 24 h period, followed by centrifugation. This procedure was repeated three times. The humic acid residue was treated with distilled water until a zero reaction to chloride was achieved [50].

The carbon (nitrogen) content of humic acids (C(N)HAs) and carbon (nitrogen) of humins (C(N)h) were calculated from the difference:

$$C(N)HAs = C(N)HAs + FAs − C(N)FAs \tag{1}$$

$$C(N)h = TOC(Nt) − C(Nt)HAs + FAs − C(N)d \tag{2}$$

The fractional composition was expressed in mg kg$^{-1}$ of dry matter of soil sample and as % share of respective fractions in the TOC (Nt) pool.

The preparations of HAs were lyophilized and powdered in agate mortar. Ash content in the HA preparations was lower than 2%.

### 2.2.3. Characteristics of Humic Acids

The humic acids separated were analyzed for the following.

Elemental composition (Perkin Elmer Series II 2400 CHN analyzer). The H/C, O/C, O/H, N/C atomic ratios and ω (internal oxidation degree) were calculated; ω was calculated according to the formula:

$$ω = (2O + 3N − H):C \tag{3}$$

where O, N, H, C—content in atomic % [10].

UV-VIS absorption spectra (Perkin Elmer UV-VIS Spectrometer, Lambda 20). VIS spectra were obtained from 0.02% humic acid solutions in 0.1 M NaOH and UV-spectra after fivefold dilution. Absorbance was measured at 280 nm ($A_{280}$), 400 nm ($A_{400}$), 465 nm ($A_{465}$), 600 nm ($A_{600}$) and 665 nm ($A_{665}$) to calculate the coefficient values:

$A_{2/4}$—280 nm and 465 nm absorbance ratio;
$A_{2/6}$—280 nm and 665 nm absorbance ratio;
$A_{4/6}$—465 nm and 665 nm absorbance ratio;
$\Delta \log K = \log A_{400} − \log A_{600}$ [51].

Infrared spectra (Perkin-Elmer FT-IR Spectrometer, Spectrum BX) over 400–4400 cm$^{-1}$ were obtained for HAs (3 mg) in KBr (800 mg). Deconvolution was applied, with a filter making the bands of γ = 4 narrower, and using the process of smoothing, for which the length parameter was l = 80% [17].

Hydrophilic and hydrophobic properties were determined with HPLC Series 200 liquid chromatograph with a DAD detector by Perkin-Elmer. The separation involved the use of column X-Terra C18, 5 μm, 250 ×4.6 mm. The solutions of humic acids were applied in 0.01 mol L$^{-1}$ NaOH of the concentration of 2 mg mL$^{-1}$; injection of the sample—10 μL; solvent—acetonitrile–water; solvent flow in the gradient (ratio $H_2O$:ACN (*v/v*) over 0–6 min—99.5:0.5, 7–13 min—70:30, 13–20 min—10:90); detection—at the excitation/emission wavelength (λex/λem) 270/330 nm. Based on the areas determined under

peaks, the share of hydrophilic (HIL) and hydrophobic ($\Sigma$HOB = HOB-1 + HOB-2 + HOB-3) fractions in humic acid molecules and the parameter HIL/$\Sigma$HOB were determined [26,30].

EPR measurements were carried out at room temperature in air using a Radiopan X-band spectrometer with 100 kHz field modulation and modulation amplitude of 0.01 mT. The absolute number of spins in the samples was obtained by comparison, under the same experimental conditions, with an $\alpha$, $\alpha$-diphenyl-$\beta$-picrylhydrazyl (DPPH) reference. Before EPR measurements, the samples were weighed and placed in a spectroscopically pure quartz tube. EPR signals were recorded using microwave power of 5 mW [52].

### 2.2.4. Statistical Analyses

The results were statistically verified by determining the standard deviation. To determine the significance of differences in the parameters, the analysis of variance (ANOVA) was conducted for $p < 0.05$ [53]. The significance of the effect of the factors and interactions was verified with test F, and the significance of differences between the values of respective traits was verified with the post hoc Tukey test at $p = 0.05$. Statistical calculations were performed in three repetitions. Moreover, for the parameters determined for humic acids, principal component analysis (PCA) and cluster analysis were performed. PCA allows, among other things, the number of variables describing phenomena to be reduced, and is used to find regularities between variables. Cluster analysis divides a dataset into groups to obtain clusters that contain elements that are similar among the cluster but different from the elements in the other groups. Groups of similar treatments are presented as a dendrogram. In a given group, the smaller the Euclidean distance, the more similar the objects are. Data clustering was performed by the Ward method [54]. The analysis was performed after data standardization.

Correlations between the examined parameters were determined using Pearson's correlation coefficients ($p \leq 0.05$). The above relationships were defined using STATISTICA MS 13 statistical software.

## 3. Results and Discussion

### 3.1. Basic Parameters of Organic Matter

One of the basic indicators of soil fertility is organic carbon content (TOC) because of its part in influencing the biological, chemical and physical properties of soil and crop yields. TOC content in soil samples without additives (control) was 12.26 g kg$^{-1}$ (average value, Table 2). The highest increase in TOC content (82.3%) was recorded in the surface layer of soil covered with chopped straw. This high increase in TOC content also caused an increase in the TOC content in the 10–20 cm layer. Blanco-Canqui and Ruis [55] showed that mulching straw on the soil surface increases the carbon content in the surface layer only and does not favor sequestration. The cited authors add that the effect that tillage method has on soil properties depends largely on the duration of a given factor. The results obtained here clearly indicate the possibility of carbon sequestration by mulching.

An increase in TOC content over control was also noted for variants B (chopped straw + CaO) and G (manure mixed with straw) at 15.74% and 16.80%, respectively. By contrast, organic carbon content was lowest in the soil samples mixed only with calcium oxide (variant H). For the remaining variants, no significant differences in TOC content were noted compared to the control. The results confirm the finding of Aye et al. [56] that long-term liming (34 years) reduces TOC content. Liming enhances the OC mineralization processes due to the increase in pH in the soil. However, the authors of this study report that the drop in TOC can be compensated by introducing exogenous organic matter (EOM) into the soil, as confirmed by the increase in TOC content obtained in variant B (chopped straw + CaO).

**Table 2.** Content of total organic carbon (TOC) and total nitrogen (Nt) and content and share of dissolved organic carbon (DOC) and nitrogen (DNt).

| Sample | TOC | Nt | TOC/Nt | DOC | DOC | DNt | DNt |
|---|---|---|---|---|---|---|---|
| | g kg$^{-1}$ | | | mg kg$^{-1}$ | % | mg kg$^{-1}$ | % |
| A1 | 22.35 ± 0.61 * | 2.49 ± 0.16 [a] | 9.01 [ab] | 286.1 ± 14.1 [a] | 1.28 ± 0.06 [bc] | 53.4 ± 1.7 [a] | 2.14 ± 0.07 [ab] |
| A2 | 13.34 ± 073 [bc] | 1.14 ± 0.11 [de] | 8.75 [ab] | 165.9 ± 15.0 [bcd] | 1.24 ± 0.1 [bc] | 27.2 ± 0.4 [cd] | 2.39 ± 0.03 [a] |
| B | 14.19 ± 0.73 [b] | 1.65 ± 0.09 [b] | 8.59 [b] | 179.0 ± 12.2 [b] | 1.26 ± 0.09 [bc] | 28.8 ± 1.6 [c] | 1.74 ± 0.10 [c] |
| C | 11.77 ± 0.64 [cde] | 1.34 ± 0.25 [cd] | 8.79 [ab] | 164.5 ± 12.4 [bcd] | 1.40 ± 0.10 [b] | 17.5 ± 1.5 [f] | 1.31 ± 0.11 [d] |
| D | 11.52 ± 0.48 [de] | 1.32 ± 0.07 [cd] | 8.74 [ab] | 141.3 ± 6.8 [de] | 1.23 ± 0.06 [bc] | 17.7 ± 0.9 [f] | 1.34 ± 0.07 [d] |
| E | 10.82 ± 0.78 [ef] | 1.24 ± 0.05 [cd] | 8.71 [ab] | 124.6 ± 5.3 [e] | 1.15 ± 0.05 [c] | 27.9 ± 1.7 [c] | 2.25 ± 0.14 [a] |
| F | 12.61 ± 0.31 [bcd] | 1.48 ± 0.12 [bc] | 8.53 [b] | 151.8 ± 5.5 [bcde] | 1.20 ± 0.04 [bc] | 35.7 ± 2.3 [b] | 2.41 ± 0.15 [a] |
| G | 14.32 ± 0.33 [b] | 1.54 ± 0.19 [b] | 9.33 [ab] | 166.2 ± 4.4 [bcd] | 1.16 ± 0.03 [c] | 20.0 ± 1.1 [ef] | 1.29 ± 0.07 [d] |
| H | 9.69 ± 0.70 [f] | 1.06 ± 0.05 [e] | 9.11 [ab] | 172.4 ± 4.9 [bc] | 1.78 ± 0.05 [a] | 19.8 ± 1.0 [ef] | 1.87 ± 0.10 [bc] |
| K | 12.26 ± 0.65 [cde] | 1.31 ± 0.08 [cd] | 9.37 [a] | 148.4 ± 6.4 [cde] | 1.21 ± 0.05 [bc] | 23.6 ± 1.4 [de] | 1.80 ± 0.11 [c] |

*—values followed by a lower-case letter are not significantly different at 5 %

The nitrogen content in soil samples without additives averaged 1.31 g kg$^{-1}$. The highest Nt content was found in the soil samples of variant A1 (chopped straw on the soil surface) at 2.49 g kg$^{-1}$. For the remaining variants, the Nt content ranged from 1.06 (variant H—soil mixed with CaO) to 1.65 (variant B—soil mixed with straw and CaO). The soil samples for variants F (soil + chopped straw) and G (soil + manure) had high nitrogen content. The TOC/Nt ratio values are derived from the TOC and Nt content. In general, the TOC/Nt values were not varied, ranging from 8.53 (soil + chopped straw) to 9.37 (control). The slight differences in the values of this ratio recorded in the research confirm the well-known dependence that the TOC/Nt ratio in soils is a relatively constant value. Even introducing organic materials (straw), which initially increases this ratio, leads to the soil ultimately achieving a state characteristic of its soil type.

The content of dissolved organic carbon (DOC) ranged from 124.6 (soil mixed with UGmax) to 286.1 mg kg$^{-1}$ (soil covered with chopped straw) (Table 2) and had a strong positive correlation with TOC content (Table 3). DOC constituted from 1.15 to 1.78% of the TOC content. The soil mixed with CaO had the highest share of this fraction of organic matter. The soil mixed with straw and nitrogen addition had a high share (1.40%) of DOC, while the lowest share was in the soil with the addition of UGmax (variant E) and the soil with manure.

**Table 3.** Significant correlation coefficients ($p \leq 0.05$) between the TOC and Nt content (g/kg) and fractional composition of humus.

| | DOC | DNt | CHAs | CFAs | NHAs | NFAs | CHAs | CFAs | Ch | NFAs | Nh |
|---|---|---|---|---|---|---|---|---|---|---|---|
| | mg kg$^{-1}$ | | | | | | % | | | | |
| TOC | 0.908 | 0.835 | 0.676 | 0.961 | 0.861 | 0.849 | - | −0.778 | 0.726 | −0.793 | 0.552 |
| Nt | 0.870 | 0.827 | 0.567 | 0.881 | 0.746 | 0.800 | −0.629 | −0.797 | 0.763 | −0.917 | 0.756 |

The nitrogen content in dissolved organic matter for variant A1 (soil + chopped straw left on the soil surface) was 53.4 mg kg$^{-1}$, which constituted 2.14% Nt. The DNt content for the remaining variants ranged from 17.5 (soil + chopped straw + N) to 35.7 (soil + chopped straw) and was correlated positively with the Nt content (Table 3). The above range of DNt contents was from 1.31 to 2.41% Nt.

An important role in shaping the properties of organic matter is played by the content and fraction of humic acids, fulvic acids and humins. In the process of fractionating organic matter, the first stage is decalcification. Carbon content in the solutions after decalcification (Cd) ranged from 194.6 to 280.5 mg kg$^{-1}$, which was 1.17 to 2.79% of the TOC (Table 4).

The lowest content of carbon fractions of fulvic acids (CFAs) was found in the soil samples without additives (variant K) and the soil mixed with CaO (variant H). The share of CFAs in these samples was 13.28 and 16.56% of TOC, respectively. The soil surface layer of variant A had the highest content of CFAs while also having the lowest share (CFAs constituting 12.34% of TOC). The samples of variants A2 and G (soil mixed with manure) also had high contents of CFAs. It should be emphasized that content of CFAs correlated positively with content of TOC, while share of CFAs correlated negatively with TOC (Table 3). The carbon fraction content in the humic acid (CHAs) was generally lower than that of the fraction in the fulvic acid. The content of CHAs was highest for the soil mixed with manure, and lowest for the soil without additives and the soil mixed with straw and UGmax. The content of CHAs—similarly to CFAs—correlated positively with carbon content (Table 3). However, no significant relationships were found between share of carbon fraction and TOC content. It should be kept in mind that the quality of organic matter is determined by the share of individual organic matter fractions. The share of CHAs was lowest in the soil to which chopped straw was applied (variant A1). The share of CHAs in the remaining variants ranged from 10.43 (soil without additives) to 14.92% (soil mixed with manure).

**Table 4.** Content and share of carbon in humus fraction.

| Sample | Cd | CFAs | CHAs | CHAs/ CFAs | Cd | CFAs | CHAs | Ch |
|---|---|---|---|---|---|---|---|---|
| | mg kg$^{-1}$ | | | | % of TOC | | | |
| A1 | 260.8 ± 5.1 [ab*] | 2757.5 ± 58.5 [a] | 1948.5 ± 32.3 [b] | 0.71 [e] | 1.17 ± 0.02 [f] | 12.34 ± 0.26 [g] | 8.72 ± 0.14 [f] | 77.78 ± 0.37 [a] |
| A2 | 219.4 ± 7.9 [cde] | 2065.0 ± 21.3 [bc] | 1751.0 ± 25.8 [c] | 0.85 [bc] | 1.64 ± 0.06 [cd] | 15.48 ± 0.16 [bc] | 13.13 ± 0.19 [b] | 69.75 ± 0.08 [e] |
| B | 202.9 ± 12.3 [ef] | 2013.1 ± 15.9 [c] | 1629.9 ± 26.9 [d] | 0.81 [cd] | 1.43 ± 0.09 [e] | 14.19 ± 0.11 [e] | 11.49 ± 0.19 [d] | 72.90 ± 0.21 [b] |
| C | 243.0 ± 7.2b [ce] | 1715.6 ± 35.4 [e] | 1395.4 ± 13.3 [e] | 0.81 [cd] | 2.06 ± 0.06 [b] | 14.57 ± 0.30 [de] | 11.86 ± 0.11 [d] | 71.50 ± 0.31 [cd] |
| D | 217.3 ± 3.1 [df] | 1743.1 ± 25.0 [e] | 1249.9 ± 14.5 [f] | 0.72 [e] | 1.89 ± 0.03 [bc] | 15.13 ± 0.22 [cd] | 10.84 ± 0.13 [e] | 72.13 ± 0.15 [bc] |
| E | 280.5 ± 8.3 [a] | 1743.1 ± 33.4 [e] | 1355.4 ± 23.0 [e] | 0.78 [de] | 2.59 ± 0.08 [a] | 16.10 ± 0.31 [ab] | 12.53 ± 0.21 [c] | 68.78 ± 0.59 [f] |
| F | 194.6 ± 7.1 [f] | 1848.8 ± 12.5 [d] | 1603.3 ± 19.4 [d] | 0.87 [b] | 1.54 ± 0.06 [de] | 14.66 ± 0.10d [e] | 12.71 ± 0.15 [bc] | 71.08 ± 0.29 [d] |
| G | 233.1 ± 2.9 [cd] | 2148.1 ± 27.3 [b] | 2135.9 ± 10.0 [a] | 0.99 [a] | 1.63 ± 0.02 [de] | 15.00 ± 0.19 [cd] | 14.92 ± 0.07 [a] | 68.45 ± 0.27 [f] |
| H | 270.8 ± 15.7 [a] | 1605.0 ± 22.8 [f] | 1403.5 ± 20.8 [e] | 0.87 [b] | 2.79 ± 0.16 [a] | 16.56 ± 0.23 [a] | 14.48 ± 0.21 [a] | 66.16 ± 0.29 [g] |
| K | 210.7 ± 4.2d [ef] | 1628.1 ± 20.7 [f] | 1278.9 ± 11.6 [f] | 0.79 [d] | 1.71 ± 0.03 [cd] | 13.28 ± 0.17 [f] | 10.43 ± 0.09 [e] | 74.57 ± 0.22 [a] |

Explanations: Cd—carbon in solutions after decalcification, CHAs—carbon of the fraction of humic acids, CFAs—carbon of the fraction of fulvic acids, Ch—carbon of the humin fraction, *—values followed by a lower-case letter are not significantly different at 5 %

The values of the CHAs/CFAs ratio are derived from the carbon content of humic and fulvic acids. It is widely assumed that humus with higher values of this ratio is typical of more fertile soils with more humified organic matter [12,23]. The values of the CHAs/CFAs ratio ranged from 0.71 (variant A1, sample with a very low degree of humification) to 0.99 (soil mixed with manure) (Table 4). Soil samples with the addition of UGmax had CHAs/CFAs values as low as in variant A1 soil. Cao et al. [12] report that low values of CHAs/CFAs may indicate high microbial activity in soils, promoting the formation of fulvic acids, which explains the low values of CHAs/CFAs for soil samples with the addition of UGmax. In addition, research by Debska et al. [13] and Piotrowska et al. [48] shows that introducing UGmax into the soil significantly increases the activity of cellulase—the group of enzymes that participate in cellulose decomposition. On the one hand, the increase in enzymatic activity may indicate the intensification of mineralization processes, while, on the other hand, increased activity gives rise to humification. According to reports in the literature [10,57], one humification mechanism is polycondensation and polymerization of simpler, lower-molecular-weight compounds formed during biochemical transformations of macromolecules (cellulose and lignin), leading firstly to the formation of fulvic acids.

The fraction of organic matter most resistant to decay is that of humins, and their participation is important in the process of carbon sequestration [11]. The share of the humin (Ch) fraction in the analyzed soil samples was high, ranging from 66.16 (soil mixed with CaO) to 77.78% (soil covered with chopped straw, variant A1) (Table 4). With the

exception of variant A1, there was a slight decrease in the share of humins compared to the control, ranging from approx. 2 to approx. 8%.

The nitrogen content in the solutions after decalcification (Nd) ranged from 1.33 to 2.71% Nt (Table 5). The nitrogen content in the fulvic acid fraction ranged from 131.6 to 167.6 g kg$^{-1}$, and the nitrogen content in the humic acid fraction (NHAs) ranged from 119.5 (soil without additives) to 221.4 g kg$^{-1}$ (soil covered with straw). Despite the content of NHAs being so high, the soil of variant A1 had the lowest share of NHAs. The values of the NHAs/NFAs ratio are derived from the nitrogen content in the humic and fulvic acid fractions. The NHAs/NFAs ratio was highest for the soil of variant A, and lowest for the soil without additives (0.82). A high value of this ratio was obtained for soil mixed with manure (1.21). For the remaining variants, NHAs/NFAs ranged from 0.9 to 1.09. The share of nitrogen in the humin fraction ranged from 69.32 to 82.89% Nt and was correlated positively with the nitrogen content (Table 3).

**Table 5.** Content and share of nitrogen in humus fraction.

| Sample | Nd | NFAs | NHAs | NHAs/ NFAs | Nd | NFAs | NHAs | Nh |
|---|---|---|---|---|---|---|---|---|
| | mg kg$^{-1}$ | | | | % of Nt | | | |
| A1 | 37.0 ± 2.7 [a*] | 167.6 ± 2.4 [a] | 221.4 ± 7.6 [a] | 1.32 [a] | 1.49 ± 0.11 [d] | 8.89 ± 0.31 [e] | 6.73 ± 0.07 [d] | 82.89 ± 0.32 [a] |
| A2 | 23.8 ± 2.6 [cd] | 143.3 ± 6.3 [cd] | 182.7 ± 4.6 [b] | 1.27 [a] | 2.09 ± 0.23 [bc] | 16.03 ± 0.41 [a] | 12.57 ± 0.55 [a] | 69.32 ± 0.73 [f] |
| B | 27.7 ± 2.2 [bc] | 151.4 ± 7.3 [abc] | 165.6 ± 7.6 [c] | 1.09 [bc] | 1.68 ± 0.13 [cd] | 10.04 ± 0.46 [cd] | 9.18 ± 0.44 [c] | 79.11 ± 0.18 [b] |
| C | 18.8 ± 1.6 [d] | 131.6 ± 5.6 [d] | 136.9 ± 3.1 [d] | 1.04 [cd] | 1.40 ± 0.13 [d] | 10.21 ± 0.23 [c] | 9.82 ± 0.41 [bc] | 78.56 ± 0.70 [b] |
| D | 20.7 ± 2.1 [d] | 136.4 ± 6.0 [cd] | 129.1 ± 5.4 [de] | 0.95 [def] | 1.57 ± 0.16 [d] | 9.78 ± 0.41 [cde] | 10.33 ± 0.46 [bc] | 78.56 ± 0.74 [b] |
| E | 33.6 ± 2.9 [ab] | 134.2 ± 5.2 [d] | 129.8 ± 2.5 [de] | 0.97 [cde] | 2.71 ± 0.24 [a] | 10.47 ± 0.21 [c] | 10.82 ± 0.42 [b] | 76.00 ± 0.82 [cd] |
| F | 33.7 ± 4.1 [ab] | 145.1 ± 6.3 [bcd] | 155.9 ± 5.7 [c] | 1.07 [bcd] | 2.28 ± 0.28 [ab] | 10.53 ± 0.38 [c] | 9.80 ± 0.43 [bc] | 77.38 ± 0.73 [bc] |
| G | 20.5 ± 1.5 [d] | 161.9 ± 7.6 [ab] | 195.6 ± 4.2 [b] | 1.21 [ab] | 1.33 ± 0.10 [d] | 12.70 ± 0.28 [b] | 10.51 ± 0.50 [b] | 75.45 ± 0.73 [d] |
| H | 23.0 ± 1.4 [cd] | 137.8 ± 6.8 [cd] | 123.7 ± 5.9 [de] | 0.90 [ef] | 2.17 ± 0.13 [bc] | 11.67 ± 0.56 [b] | 13.00 ± 0.64 [a] | 73.16 ± 1.13 [e] |
| K | 31.0 ± 1.3 [ab] | 145.0 ± 3.0 [bcd] | 119.5 ± 2.2 [e] | 0.82 [f] | 2.37 ± 0.10 [ab] | 9.12 ± 0.17 [de] | 11.07 ± 0.23 [b] | 77.44 ± 0.15 [bc] |

Explanations: Nd—nitrogen in solutions after decalcification, NHAs—nitrogen of the fraction of humic acids, NFAs—nitrogen of the fraction of fulvic acids, Nh—nitrogen of the humin fraction, *—values followed by a lower-case letter are not significantly different at 5 %

### 3.2. Properties of Humic Acids

To determine the role and importance of organic matter in the soil environment, apart from determining the content of various OM fractions, it is important to know the properties of OM. In this study, the properties of HAs, which constitute one of the most important OM fractions, were isolated and determined. Due to the complex structure of humic acids and the variability of their properties according to the conditions in which they arise, no single analytical tool is sufficient to determine changes in the properties of humic acids under the influence of anthropogenic factors, e.g., [58]. In this study, the properties of humic acids were determined based on elemental composition, spectrometric properties in the UV-VIS range, infrared spectra, hydrophilic–hydrophobic properties (HPLC) and EPR measurements.

#### 3.2.1. Elemental Composition of Humic Acids

The elemental composition of humic acids (HAs) expressed in atomic % is presented in Table 6. The HAs of the variants with UGmax had the highest carbon content and the lowest hydrogen content. The oxygen content in the analyzed samples (D and E) differed significantly only from the HAs of variant B (chopped straw + CaO mixed with soil). In the other variants, the carbon content ranged from 36.65 to 37.93%, hydrogen from 39.45 to 40.37% and oxygen from 18.75 (variant B) to 20.46% (variant C). The nitrogen content in the analyzed HAs ranged from 2.98 to 3.10%.

**Table 6.** Elemental composition and atomic ratio of humic acids.

| Sample | C | H | N | O | H/C | N/C | O/C | O/H | ω |
|---|---|---|---|---|---|---|---|---|---|
| A1 | 36.65 ± 0.01 f* | 40.27 ± 0.08 a | 3.10 ± 0.09 a | 19.98 ± 0.44 ab | 1.10 ± 0.01 a | 0.085 ± 0.002 ab | 0.545 ± 0.012 a | 0.496 ± 0.012 cd | 0.245 ± 0.026 abcd |
| A2 | 37.39 ± 0.17d e | 40.36 ± 0.32 a | 3.09 ± 0.05 a | 19.16 ± 0.56 ab | 1.08 ± 0.01 ab | 0.083 ± 0.002 ab | 0.512 ± 0.013 ab | 0.475 ± 0.017 d | 0.193 ± 0.036 cd |
| B | 37.84 ± 0.10 bc | 40.37 ± 0.32 a | 3.06 ± 0.05 a | 18.75 ± 0.40 b | 1.07 ± 0.01 bc | 0.081 ± 0.001 abc | 0.495 ± 0.010 b | 0.464 ± 0.013 d | 0.167 ± 0.025 d |
| C | 37.24 ± 0.07 e | 39.34 ± 0.30 b | 2.98 ± 0.09 a | 20.46 ± 0.68 a | 1.06 ± 0.01b cd | 0.080 ± 0.002 abc | 0.549 ± 0.017 a | 0.520 ± 0.020 abc | 0.282 ± 0.049 abc |
| D | 39.39 ± 0.02 a | 36.92 ± 0.19 c | 3.02 ± 0.04 a | 20.69 ± 0.37 a | 0.94 ± 0.01 f | 0.077 ± 0.001 c | 0.525 ± 0.009 ab | 0.560 ± 0.012 a | 0.344 ± 0.025 a |
| E | 38.97 ± 0.10 a | 37.62 ± 0.16 c | 3.04 ± 0.03 a | 20.38 ± 0.42 a | 0.97 ± 0.01 e | 0.078 ± 0.001 bc | 0.523 ± 0.012 ab | 0.542 ± 0.0093 ab | 0.315 ± 0.021 ab |
| F | 37.93 ± 0.11 b | 39.45 ± 0.20 b | 3.06 ± 0.04 a | 19.57 ± 0.81 ab | 1.04 ± 0.01 d | 0.081 ± 0.001 abc | 0.516 ± 0.023 ab | 0.496 ± 0.018 cd | 0.234 ± 0.040 bcd |
| G | 37.80 ± 0.13 bc | 39.71 ± 0.33 ab | 3.03 ± 0.04 a | 19.47 ± 0.64 ab | 1.05 ± 0.01 cd | 0.080 ± 0.001 abc | 0.515 ± 0.016 ab | 0.490 ± 0.020 cd | 0.220 ± 0.038 bcd |
| H | 37.60 ± 0.08 c | 39.54 ± 0.16 b | 2.98 ± 0.07 a | 19.89 ± 0.60 ab | 1.05 ± 0.01 cd | 0.079 ± 0.002 bc | 0.529 ± 0.017 ab | 0.503 ± 0.013 bcd | 0.244 ± 0.034 abcd |
| K | 37.15 ± 0.06 de | 39.81 ± 0.11 ab | 2.99 ± 0.09 a | 20.06 ± 0.70 ab | 1.07 ± 0.01 bc | 0.080 ± 0.002 abc | 0.540 ± 0.019 a | 0.504 ± 0.016 bcd | 0.250 ± 0.039 abcd |

*—values followed by a lower-case letter are not significantly different at 5 %.

The quality (properties) of HAs are significantly indicated by the atomic ratios of individual elements of the HAs (Table 6). High carbon and oxygen content and low hydrogen content and, consequently, low values of the H/C ratio and high degrees of internal oxidation and atomic ratios of O/C and O/H testify to a high maturity of HA molecules. This maturity is associated with a high level of soil organic matter humification [10,26,31,34,37,59–61]. Undoubtedly, the variants with UGmax (variant D) showed the highest degree of internal oxidation (ω) and O/H. Variants D and E also exhibited the lowest H/C ratios (0.94 and 0.97). The humic acids of the remaining variants were characterized by H/C ratios ranging from 1.04 to 1.10. Generally, the results of the HAs' elemental compositions indicate a high degree of humification of organic matter in the tested soil samples, regardless of type of fertilizer (EOM). The parameters of HA quality are generally similar to the control HAs (the exception being the variant with UGMax). According to Zavyalova [62], the HAs of soils with no influx of fresh organic matter should be characterized by a high degree of HA maturity and/or organic matter humification.

### 3.2.2. Spectrometric Parameters in the UV-VIS Range

According to the literature [32,34,35,50], spectrometric parameters (absorbance and coefficients of absorbance) are important indicators of degree of maturity HAs. The $A_{2/4}$ ratio describes the ratio of content of decomposition-resistant lignins to poorly humified organic matter; $A_{2/6}$ describes the ratio of content of decomposition-resistant lignin compounds to highly humified organic matter. HAs with a higher degree of "maturity" (as compared to HAs with a lower molecular weight, and thus lower degree of "maturity") show lower absorbance values and $A_{4/6}$ and ΔlogK coefficients. Kumada [50], based on spectrometric parameters in the UV-VIS range, divided HAs into three basic types: type A are those with a high degree of humification, for which ΔlogK reaches values of up to 0.6; type B have ΔlogK values from 0.6 to 0.8; and type Rp have coefficient values from 0.8 to 1.1.

The lowest absorbance values in the UV-VIS range as a whole were found in the HAs of the variant in which the organic material was not mixed in with the soil (A1) (Table 7). The largest difference in absorbance values was between HAs of variant A1 and those of variants D, E (soil with addition of UGmax) and H (soil with addition of CaO). The $A_{2/4}$ ratio ranged from 5.60 (variant A1) to 7.29 (variant F); the $A_{2/6}$ ratio ranged from 29.5 for the HAs of the soil without additives to 47.2 for the HAs of the soil mixed with manure (Table 7). The $A_{4/6}$ ratio ranged from 5.08 for the HAs of soil without additives to 6.73 for the HAs of soil mixed with straw with the addition of nitrogen. The values of the $A_{4/6}$ and $A_{2/6}$ parameters clearly indicate that the HAs of the soil without additives had the highest degree of maturity. The $A_{4/6}$ values for HAs ranged from 5.73 to 5.99 for the variants with the addition of UGmax (D, E) and CaO (variant H 5.82), and for the remaining variants ranged from 6.10 to 6.73 (variants with chopped straw and manure). In general, according to the division presented by Kumada [50], all analyzed HAs can be classified as one type (type B). Thus, introducing neither straw nor manure into the soil, nor applying UGmax or additional nitrogen fertilization, nor liming drastically changes the properties of organic

matter. This is important for maintaining the equilibrium state characteristic of a given soil type.

**Table 7.** Absorbance values and coefficients of absorbance of humic acids.

| Sample | $A_{280}$ | $A_{400}$ | $A_{465}$ | $A_{600}$ | $A_{665}$ | $A_{2/4}$ | $A_{2/6}$ | $A_{4/6}$ | $\Delta logK$ ** |
|---|---|---|---|---|---|---|---|---|---|
| A1 | 4.00 ± 0.17 * | 1.31 ± 0.03 | 0.714 ± 0.003 | 0.256 ± 0.005 | 0.117 ± 0.003 | 5.60 ± 0.26 [f] | 34.2 ± 1.79 [e] | 6.11 ± 0.16 [bc] | 0.709 ± 0.009 [ab] |
| A2 | 6.61 ± 0.10 | 1.67 ± 0.04 | 0.925 ± 0.004 | 0.341 ± 0.004 | 0.145 ± 0.004 | 7.15 ± 0.07 [a] | 45.6 ± 0.69 [ab] | 6.36 ± 0.16 [abc] | 0.690 ± 0.016 [abc] |
| B | 5.81 ± 0.16 | 1.51 ± 0.04 | 0.829 ± 0.006 | 0.298 ± 0.006 | 0.136 ± 0.003 | 7.01 ± 0.18 [ab] | 42.7 ± 0.41 [bc] | 6.10 ± 0.13 [bc] | 0.705 ± 0.011 [ab] |
| C | 5.31 ± 0.12 | 1.62 ± 0.04 | 0.901 ± 0.006 | 0.331 ± 0.004 | 0.134 ± 0.005 | 5.89 ± 0.17 [def] | 39.7 ± 1.95 [cd] | 6.73 ± 0.24 [a] | 0.690 ± 0.013 [abc] |
| D | 7.28 ± 0.06 | 1.87 ± 0.03 | 1.104 ± 0.005 | 0.447 ± 0.004 | 0.193 ± 0.010 | 6.60 ± 0.08 [bc] | 37.8 ± 1.78 [de] | 5.73 ± 0.31 [d] | 0.622 ± 0.010 [d] |
| E | 6.63 ± 0.17 | 1.80 ± 0.05 | 1.048 ± 0.030 | 0.413 ± 0.011 | 0.175 ± 0.004 | 6.33 ± 0.05 [cd] | 37.9 ± 0.53d [e] | 5.99 ± 0.05 [cd] | 0.639 ± 0.004 [d] |
| F | 6.59 ± 0.15 | 1.59 ± 0.04 | 0.904 ± 0.005 | 0.345 ± 0.018 | 0.148 ± 0.004 | 7.29 ± 0.13 [a] | 44.5 ± 0.15 [ab] | 6.10 ± 0.11 [bc] | 0.664 ± 0.029 [cd] |
| G | 6.09 ± 0.21 | 1.57 ± 0.04 | 0.847 ± 0.007 | 0.302 ± 0.017 | 0.129 ± 0.004 | 7.19 ± 0.23 [a] | 47.2 ± 2.64 [a] | 6.57 ± 0.24 [ab] | 0.716 ± 0.029 [a] |
| H | 6.87 ± 0.08 | 1.85 ± 0.04 | 1.11 ± 0.040 | 0.423 ± 0.020 | 0.191 ± 0.008 | 6.19 ± 0.16 [cde] | 36.0 ± 1.09d [e] | 5.82 ± 0.08 [d] | 0.642 ± 0.018 [cd] |
| K | 5.40 ± 0.13 | 1.56 ± 0.05 | 0.928 ± 0.007 | 0.349 ± 0.011 | 0.183 ± 0.006 | 5.82 ± 0.18 [ef] | 29.5 ± 0.29 [f] | 5.08 ± 0.20 [e] | 0.650 ± 0.008 [cd] |

*—values followed by a lower-case letter are not significantly different at 5 %, ** $\Delta logK = log\ A_{400} - log\ A_{600}$.

### 3.2.3. Hydrophilic–Hydrophobic Properties of Humic Acids

An exemplary RP-HPLC chromatogram of humic acids is shown in Figure 1. Based on the course of the chromatograms, one hydrophilic fraction and three time intervals in which the hydrophobic fractions occurred were distinguished: HIL—3.94–4.47 min; HOB-1—13.57–15.72 min; HOB-2—16.18–19.06 min; HOB-3—19.32–23.37 min.

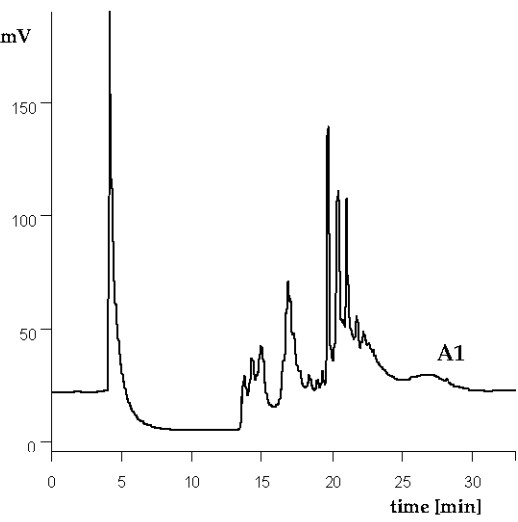

**Figure 1.** Selected RP-HPLC chromatogram of humic acids.

The humic acids were distinguished into individual fractions based on differences in hydrophobicity. Fractions ranging from 3.94 to 4.47 min show greater hydrophilic properties, while fractions ranging from 13.57 to 23.37 min are mainly characterized by hydrophobic properties [63,64]. It should be remembered that the hydrophobicity of the fraction increases with increasing retention time.

The share of hydrophilic fractions ranged from 16.45 (variant E) to 22.27% (variant A1); HOB-1 ranged from 11.20 (variant B) to 17.31% (HAs of the soil without additives, variant K); HOB-2 ranged from 12.73 (variant C) to 19.34% (variant A1); and HOB-3 ranged from 43.40 (variant A1) to 55.34% (variant E) (Table 8). Thus, the hydrophobic fractions were dominated by the fraction with the longest retention time. It should also be emphasized that the HAs of the UGMax variants had the highest share of HOB-3 fraction. The HAs of these variants had the highest carbon content and lowest hydrogen content (Table 6). As shown in results from the correlations presented in Table 9 of the hydrophobic

fractions, only the HOB-3 fraction correlated significantly with carbon content (positively) and hydrogen content (negatively). Song et al. [41] emphasize that the greater the share of hydrophobic fractions, the greater the stability of HA molecules and the more intense the carbon sequestration. Trubetskaya et al. [33] reported that HAs with a higher share of hydrophobic fractions show a higher molecular weight. Kumada [65] showed that the share of hydrophobic fractions in humic acid molecules increases with increased maturity of HAs. Thus, as reported by Kumada [65] and Trubetskaya et al. [33], HAs with a higher degree of "maturity" have lower values of the HIL/ΣHOB ratio. This ratio was highest for HAs of variant A1, while significantly lower values were recorded for, among others, HAs isolated from the soil where UGmax was used. The values of the HIL/ΣHOB ratio correlated significantly with, inter alia, the H/C ratio and ΔlogK (both positively), and (negatively) with O/H and the S parameter (Table 9).

**Table 8.** Share of hydrophilic (HIL) and hydrophobic (HOB) fraction in humic acids and parameter of EPR.

| Sample | HIL | HOB-1 | HOB-2 | HOB-3 | ∑HOB | HIL/∑HOB | S ** |
|---|---|---|---|---|---|---|---|
| A1 | 22.27 ± 1.25 [a*] | 14.99 ± 0.85 [bcd] | 19.34 ± 0.69 [a] | 43.40 ± 1.35 [e] | 77.73 ± 2.60 [c] | 0.286 ± 0.008 [a] | 2.10 [d] |
| A2 | 19.96 ± 1.01 [abc] | 13.58 ± 0.52 [de] | 19.09 ± 0.47 [a] | 47.36 ± 0.89[d] | 80.04 ± 1.24 [abc] | 0.250 ± 0.015 [bcd] | 4.05 [b] |
| B | 21.02 ± 1.00 [ab] | 11.20 ± 0.89 [f] | 16.23 ± 0.41 [b] | 51.56 ± 1.23 [b] | 78.98 ± 1.40 [bc] | 0.266 ± 0.015 [b] | 2.85 [cd] |
| C | 19.06 ± 0.74b [cd] | 16.20 ± 0.75 [ab] | 12.73 ± 0.63 [c] | 52.01 ± 0.57 [b] | 80.94 ± 0.55 [abc] | 0.235 ± 0.010 [cd] | 3.40 [c] |
| D | 17.03 ± 1.29 [d] | 13.11 ± 0.72 [e] | 14.74 ± 0.67 [b] | 55.12 ± 0.71 [a] | 82.97 ± 2.04 [ab] | 0.205 ± 0.011 [ef] | 5.90 [a] |
| E | 16.45 ± 0.51 [d] | 13.13 ± 0.35 [e] | 15.08 ± 0.48 [b] | 55.34 ± 0.81 [a] | 83.55 ± 1.47 [a] | 0.197 ± 0.004 [f] | 4.30 [b] |
| F | 16.84 ± 0.77 [d] | 15.16 ± 0.53 [bcd] | 16.37 ± 0.56 [b] | 51.63 ± 0.77 [b] | 83.16 ± 1.69 [ab] | 0.202 ± 0.006 [ef] | 3.70 [c] |
| G | 20.27 ± 1.12 [abc] | 14.02 ± 0.87 [cd] | 15.21 ± 0.56 [b] | 50.51 ± 0.43 [bc] | 79.73 ± 0.86 [abc] | 0.254 ± 0.016 [bc] | 0.60 [e] |
| H | 18.26 ± 0.70 [cd] | 15.65 ± 0.83 [abc] | 15.70 ± 0.62 [b] | 50.40 ± 1.01 [bc] | 81.74 ± 0.58 [abc] | 0.223 ± 0.010d [e] | 4.50 [b] |
| K | 18.73 ± 0.75 [bcd] | 17.31 ± 0.52 [a] | 14.97 ± 0.68 [b] | 48.99 ± 0.69 [cd] | 81.27 ± 1.85 [abc] | 0.230 ± 0.004 [cd] | 3.70 [c] |

*—values followed by a lower-case letter are not significantly different at 5 %, ** [Spin g$^{-1}$] × 10$^{17}$.

**Table 9.** Significant correlation coefficients ($p \leq 0.05$) between the parameters of humic acids.

| | C | H | H/C | O/H | A$_{280}$ | A$_{465}$ | A$_{665}$ | ΔlogK | HIL | HOB-3 | ∑HOB | HIL/∑HOB |
|---|---|---|---|---|---|---|---|---|---|---|---|---|
| H | −0.880 | - | - | - | - | - | - | - | - | | | |
| H/C | −0.961 | 0.975 | - | - | - | - | - | - | - | | | |
| O/H | 0.661 | −0.938 | −0.840 | - | - | - | - | - | - | | | |
| A$_{280}$ | 0.764 | −0.557 | −0.687 | - | - | - | - | - | - | | | |
| A$_{465}$ | 0.683 | −0.709 | −0.737 | 0.635 | 0.826 | - | - | - | - | | | |
| A$_{665}$ | 0.560 | −0.621 | −0.633 | 0.589 | 0.650 | 0.905 | - | - | - | | | |
| ΔlogK | −0.630 | 0.751 | −0.734 | - | −0.623 | −0.883 | −0.945 | - | - | | | |
| HIL | −0.713 | 0.759 | 0.779 | −0.681 | −0.731 | −0.805 | −0.731 | 0.853 | - | | | |
| HOB-3 | 0.868 | −0.781 | −0.847 | 0.610 | 0.700 | 0.685 | 0.529 | −0.590 | −0.780 | | | |
| ∑HOB | 0.713 | −0.759 | −0.779 | 0.681 | 0.731 | 0.805 | 0.731 | −0.853 | −0.999 | 0.780 | | |
| HIL/∑HOB | −0.705 | 0.752 | 0.772 | −0.677 | −0.731 | −0.808 | −0.735 | 0.852 | 0.999 | −0.781 | −0.999 | |
| S | 0.547 | −0.621 | −0.609 | 0.592 | 0.602 | 0.797 | 0.790 | −0.871 | −0.689 | - | 0.689 | −0.686 |

### 3.2.4. EPR Spectroscopy

Detailed examinations of humic substances revealed the presence of paramagnetic centers associated with various chemical environments, which were characterized by almost the same g-factors of 2.0024–2.0026 but with different line widths ΔBpp [66–69] (Figure 2). This seems to be confirmed by studies using EPR spectroscopy. Two different types of paramagnetic species were discerned: (i) a double line signal (very intense for the samples after UGmax application) and (ii) a much weaker single line signal for other samples.

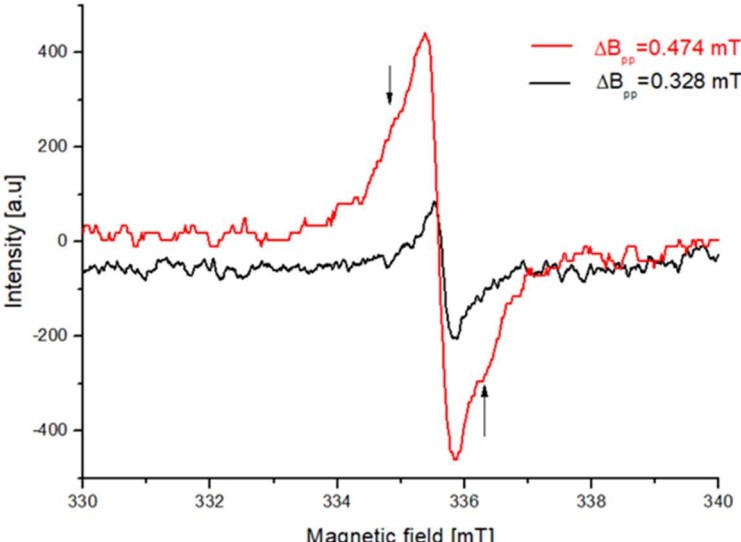

**Figure 2.** Typical EPR spectrum of the samples after UGmax application from samples D and E (red), and typical spectrum of other samples (black). The arrows indicate double character of EPR spectrum. Intense EPR line (red line) is double line i.e. consists of two signals: a strong narrower signal and a weaker wider signal (marked by arrows).

The EPR spectra with g-factors in the range of 2.0024–2.0026 can be attributed to oxygen-centered radicals or carbon-centered radicals with a nearby heteroatom, such as oxygen. The estimated concentration of spins for the samples from groups D and E is clearly higher than the concentration estimated for the remaining samples (parameter S, Table 8). It should also be noted that the line width for the samples after UGmax application is significantly greater, which may be caused by the influence of the applicator on the chemical environment of paramagnetic radicals.

As mentioned earlier, the highest values of this parameter were obtained for humic acids isolated from the soil incubated with straw and UGmax, and the lowest for the HAs of the variant with manure. The S parameter (Table 9) correlated significantly: positively with carbon content and O/H ratio and UV-VIS spectra absorbance values; and negatively with hydrogen content, the H/C ratio, ΔlogK, share of hydrophilic fractions and the HIL/ΣHOB ratio. It can therefore be assumed that the value of the S parameter increases somewhat approximately with level of maturity of humic acid molecules [70].

3.2.5. FT-IR Spectra of Humic Acids

Figure 3 shows examples of the infrared (IR) spectra of humic acids (of soil without additives and of soil incubated with straw left on its surface). The FT-IR spectra of the analyzed humic acids were characterized by the presence of the following absorption bands: $3400-3100$ cm$^{-1}$, corresponding to O–H stretching of alcohols, phenols and acids and N–H stretching; $3100-3000$ cm$^{-1}$, associated with the presence of C–H groups of aromatic and alicyclic compounds; $2960-2920$ and $2850$ cm$^{-1}$, corresponding to the asymmetric and symmetric C–H stretching of the $CH_3$ and $CH_2$ group (the intensity of these bands is taken as an indicator of the aliphaticity of HAs); $1730-1710$ cm$^{-1}$ band, indicating the presence of C=O stretching of carboxyl, aldehyde and a ketone group; $1660-1620$ cm$^{-1}$ C=O stretching of amide groups; $1610-1600$ cm$^{-1}$ indicates the presence of C–C stretching of aromatic rings; $1550-1530$ cm$^{-1}$ N–H deformation, C=N stretching (amide II bands); $1520-1500$ cm$^{-1}$ C–C stretching of aromatic rings; $1460-1440$ cm$^{-1}$ corresponds to C–H bonds—asymmetric of $CH_3$ and $CH_2$; $1420-1400$ cm$^{-1}$, C–O stretching and OH deformation of phenol groups; $1380-1320$ cm$^{-1}$, C–N aromatic amine, COO and C–H stretching; $1280-1200$ cm$^{-1}$ corresponds to C–O bond stretching of aryl ethers, esters and phenols; $1160-1030$ cm$^{-1}$, C–O stretching of alcohols, ethers and polysaccharides [32,36,40]. It should be emphasized that,

regardless of variant, the band in the 1730–1710 cm$^{-1}$ range, which is associated with the presence of C=O, was particularly intense; this indicates a high degree of oxidation of the tested humic acids (which is consistent with, among other things, the results of the elemental composition—O/C and ω). Moreover, this band exhibited the greatest differences in intensity. The intensity of the 1730–1710 cm$^{-1}$ band increased in the following order: A1 < A2 = B = G < C = F = K < H = E =D.

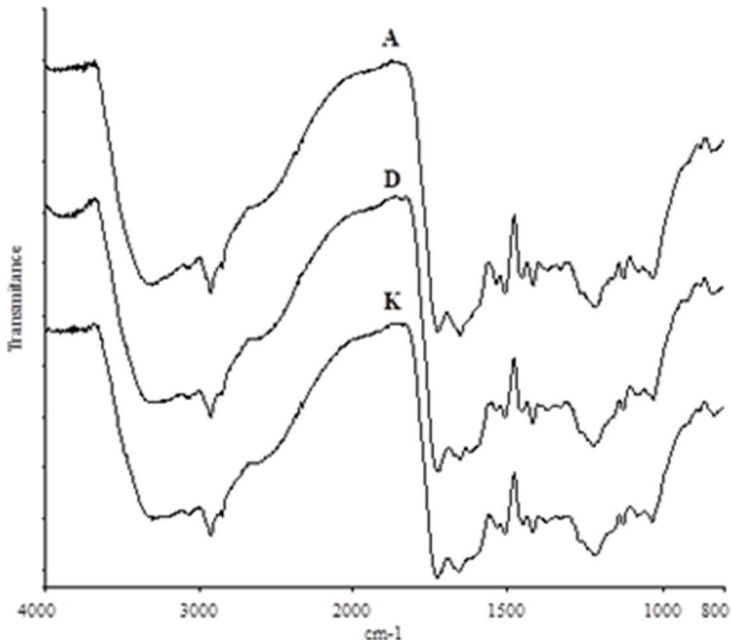

**Figure 3.** Selected FT-IR spectra of humic acids.

The course of the FT-IR spectra also exhibited slight changes in the intensity of bands in the 2960–2920 and 2850 cm$^{-1}$ ranges (the bands assumed as the aliphaticity index) and the 1160–1030 cm$^{-1}$ bands. The lowest intensity of the above-mentioned bands characterized the HAs in the soil with the addition of UGmax (variants D and E), HAs with the lowest values of the H/C ratio.

As in the literature reports [17,32,40], humic acids with higher band intensity in the range of 1730 – 1710 cm$^{-1}$ and smaller bands in the range of 2960–2920 and 2850 cm$^{-1}$ and in the range of 1160–1030 cm$^{-1}$ are characterized by a higher "degree of maturity". Thus, the FT-IR spectra indicate a higher degree of maturity of humic acids in the soil with the addition of UGmax in comparison to the HAs of the other variants.

Principal component analysis (PCA) was used to determine the variables that best describe the influence that the additives used had on HA properties. PCA serves, among other things, to reduce the number of variables describing phenomena and to detect regularities between variables. As Figure 4 shows, the first two principal components represent 76.31% of the total variance of the original dataset.

The PCA analysis showed that the first component (PCA1) accounted for 56.42% of the total variance. PCA1 was primarily positively correlated with carbon and oxygen content, the hydrophobic fraction with the longest retention time (HOB-3), the sum of hydrophobic fractions and S, as well as absorbance values in the UV-VIS range and values of the O/H and ω parameters. PCA1 was negatively correlated with hydrogen and nitrogen contents, hydrophilic fractions (HIL) and values of H/C, N/C, ΔlogK and HIL/ΣHOB parameters. PCA2 correlated positively with the content of HOB-1 and the O/C ratio, and negatively with the $A_{2/4}$ and $A_{2/6}$ ratios (Table 10).

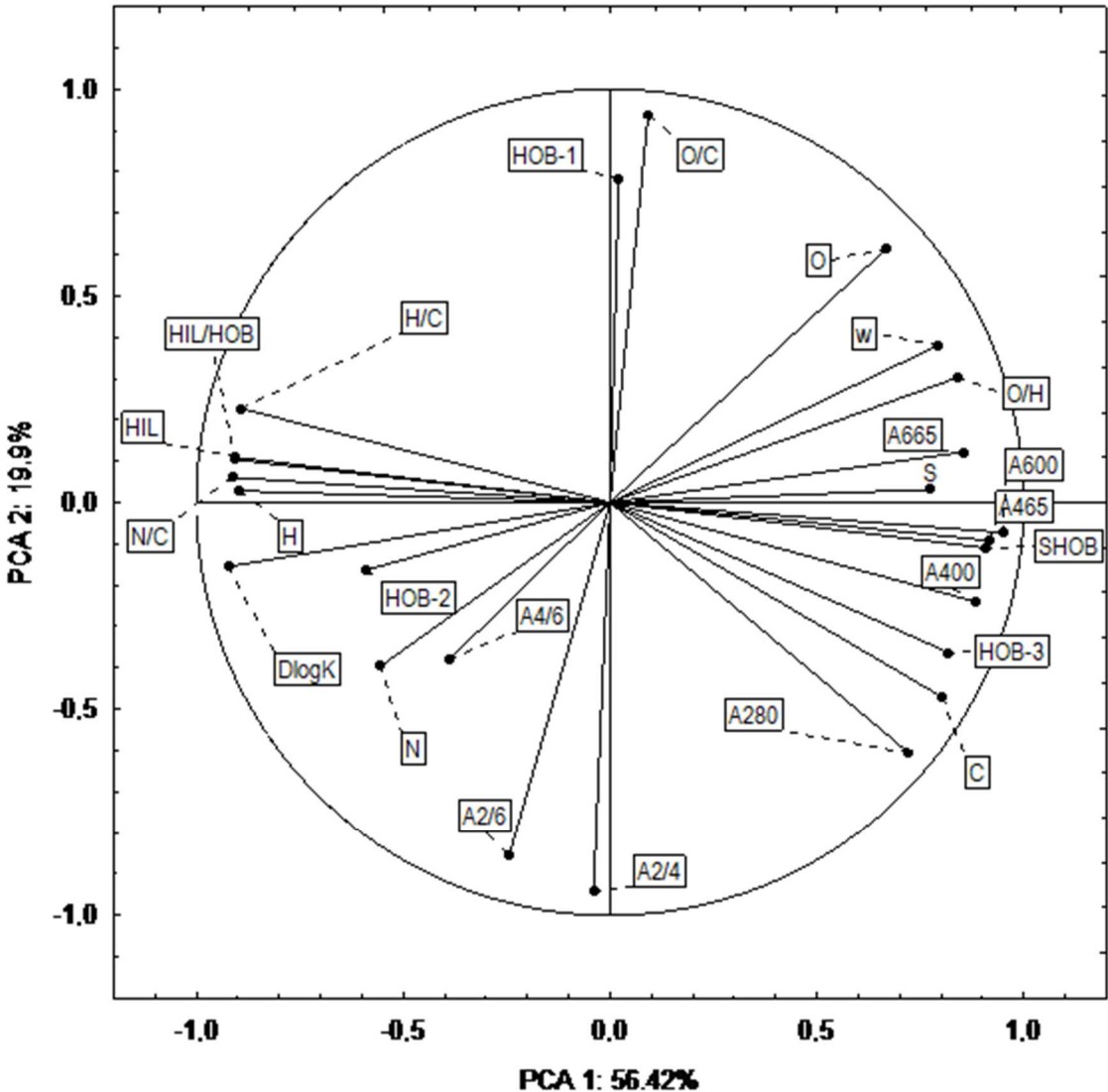

**Figure 4.** Configuration of variables in the system of the first two axes PCA1 and PCA2 of principal components.

**Table 10.** Loading scores of the variable for PCA.

| Variable | PCA1 | PCA2 | Variable | PCA1 | PCA2 |
|---|---|---|---|---|---|
| C | 0.802 | −0.470 | $A_{665}$ | 0.856 | 0.122 |
| H | −0.897 | 0.028 | $A_{2/4}$ | −0.040 | −0.942 |
| N | −0.555 | −0.394 | $A_{2/6}$ | −0.244 | −0.853 |
| O | 0.669 | 0.614 | $A_{4/6}$ | −0.389 | −0.380 |
| H/C | −0.894 | 0.228 | ΔlogK | −0.922 | −0.155 |
| N/C | −0.911 | 0.060 | HIL | −0.908 | 0.110 |
| O/C | 0.091 | 0.938 | HOB-1 | 0.021 | 0.784 |
| O/H | 0.841 | 0.303 | HOB-2 | −0.592 | −0.164 |
| ω | 0.792 | 0.380 | HOB-3 | 0.817 | −0.363 |
| $A_{280}$ | 0.720 | −0.604 | ΣHOB | 0.908 | −0.110 |
| $A_{400}$ | 0.882 | −0.239 | HIL/ΣHOB | −0.908 | 0.104 |
| $A_{600}$ | 0.951 | −0.073 | S | 0.774 | 0.032 |

The parameters correlated with the PCA1 component were used in the cluster analysis. To obtain complete information on the differences (similarities) of humic acids depending on the kind of fertilizer, the cluster analysis was applied. Humic acids with similar properties are located on dendrograms in homogenous groups.

Cluster analysis distinguished two main groups with similar properties (Figure 5):

- Group I comprises the HAs isolated from the soil samples with UGmax;
- Group II comprises the soil HAs of the remaining variants. However, this group includes subgroups that comprise:
    - Subgroup I—HAs of the soil without additives (K), the soil mixed with CaO (H) and the soil mixed with chopped straw (variants C and F);
    - Subgroup II—HAs of the soil mixed with manure, the soil from the 10–20 cm layer onto which a mulch of chopped straw was applied (A2) and the soil with chopped straw and CaO (B) and variant G (soil mixed with manure).

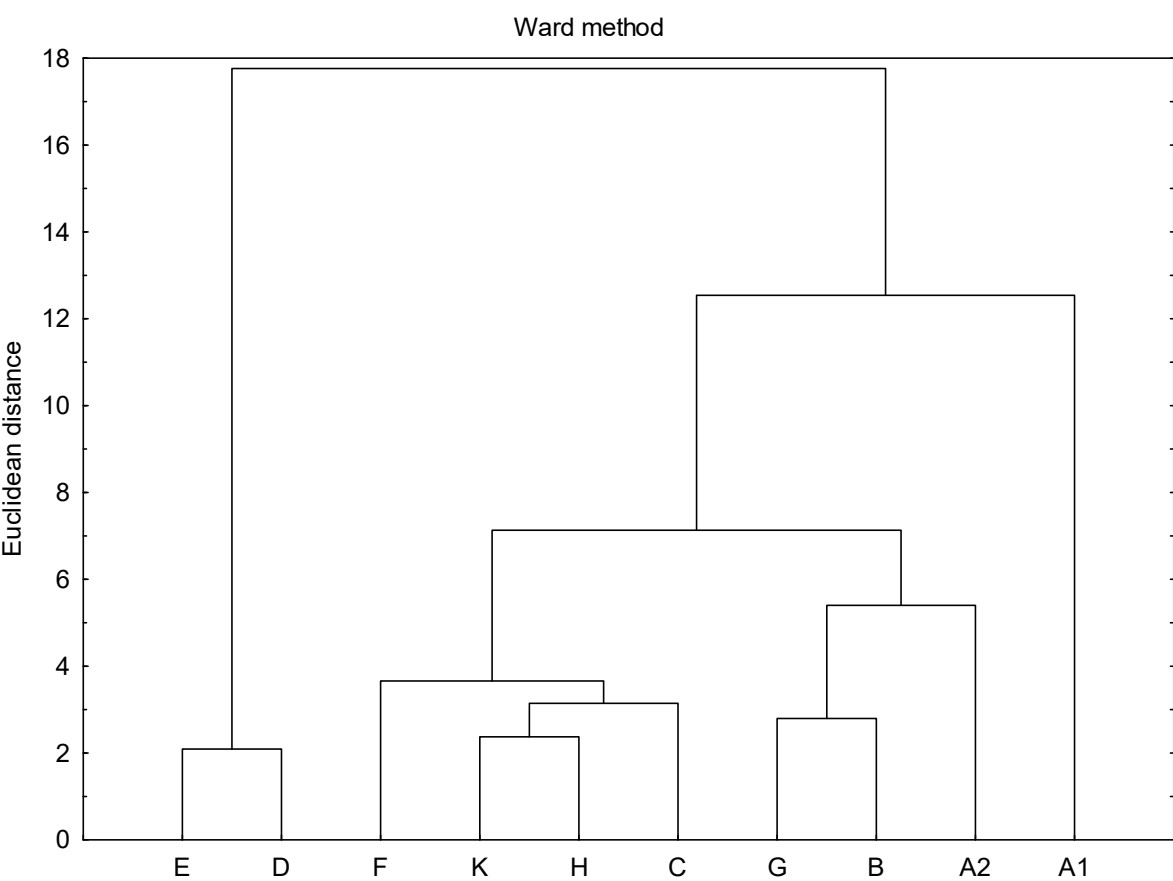

**Figure 5.** Cluster analysis determined based on humic acid parameters.

The HAs of the surface layer covered with straw mulch are lying outside the above-mentioned subgroups (variant A1). The obtained results indicate a significant role in tailoring the HA properties of the UGmax biostimulator. According to the literature review [13,43,48], the introduction of biostimulants into the soil significantly increases the enzymatic activity of soils. Enzymatic activity is one of the important factors determining the intensification of humification processes. Increase in the intensity of the organic matter humification process causes an increase in the "maturity" of HAs.

## 4. Conclusions

The effects of various types of exogenous organic matter (manure and wheat straw) were tested in a ten-year pot experiment combination with the addition of nitrogen or CaO or a biostimulant. Each of the additives contributed to shaping OM properties. However,

importantly, the degree of changes in the qualitative composition of OM and the properties of humic acids did not cause drastic changes that might affect the soil type.

The content of organic matter compared to soil without additives increased with the use of manure and the use of straw in the CaO variant and in the form of mulch. The use of a biostimulator may decrease the organic matter content in soil (by intensifying OM decomposition). However, the decrease in OM after adding the biostimulator can be limited by the addition of straw (a material that decomposes poorly).

The use of a biostimulator—with or without the addition of straw—increases carbon sequestration in humic acid molecules, their oxidation level and their share of hydrophobic fractions with the longest retention time. Thus, the addition of UGmax intensifies humification processes, leading to the formation of highly stable humic acid molecules.

**Author Contributions:** Conceptualization, B.D., K.K., M.B.-S. and E.S.-F.; Methodology, B.D., K.K., M.B.-S. and E.S.-F.; Investigation, B.D., M.B.-S. and E.S.-F.; Data curation—compilation and analysis of results, B.D., K.K., M.B.-S., E.S.-F. and E.T.; Writing—original draft, B.D., K.K., M.B.-S. and. E.S.-F.; Writing—review and editing, B.D., K.K., M.B.-S., E.S.-F. and E.T. All authors reviewed the manuscript. All authors have read and agreed to the published version of the manuscript.

**Funding:** The research has been made as part of the BN-31/2019 research project, financed by the Ministry of Science and Higher Education.

**Institutional Review Board Statement:** Not applicable.

**Informed Consent Statement:** Not applicable.

**Data Availability Statement:** Data sharing not applicable.

**Conflicts of Interest:** The authors declare no conflict of interest.

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
