# Peer review of "Soil Fertility Improvement and Carbon Sequestration through Exogenous Organic Matter and Biostimulant Application"

_agriculture, doi:10.3390/agriculture12091478_

Round 1
Reviewer 1 Report
Attached

Author Response
Dear Reviewer,
The paper's Authors wish to thank for all the precious comments and guidelines.
I hope that the corrections introduced are satisfactory. All the changes have been marked in the text.
With best regards,
Magdalena Banach-Szott
The title was changed as suggested by the Reviewer
There was: Improving soil fertility and carbon sequestration through the application of exogenous organic matter and biostimulants
There is: Soil fertility improvement and carbon sequestration through exogenous organic matter and biostimulants application
Abstract
The abstract has been corrected in line with the Reviewer's suggestions, incl. the conditions of the experiment were detailed. Was added sentences:
The research was carried out on the basis of soil samples from a ten-year pot experiment was set up as single-factor pot experiment with four replications. PVC pots with perforated bottoms were filled with soil samples taken from the tilled layer of an arable field where winter wheat was grown in monoculture. The pots were exposed directly to the weather and were left without vegetation.
Abstract shortened (all the changes made are marked in the text).
Introduction
The introduction was extended as suggested by the Reviewer and the missing literature was supplemented.
Added text:
Soil organic carbon (SOC) and nitrogen (N) are two of the most important indicators for agricultural productivity. Soil SOC and N dynamics are governed by climate change, the soil environment and human activities, mainly field management practices (Brevik 2013). The C and N cycle is important not only for improving crop efficiency, but also for mitigating climate change and the functioning of ecosystems (Law et al. 2018). From an agricultural point of view, a big challenge is to increase or maintain yields without progressive degradation of the Earth's environmental systems, especially soils (Kopittke et al. 2019). Progressing global soil degradation as a result of agricultural intensification (Hossain et al. 2020, Scherr 1999) is further exacerbated by climate change (Hari et al. 2020, Ray et al. 2019). The European Commission has developed a comprehensive green action plan to achieve the European Green Deal target of 25% of agricultural land being farmed organically by 2030 [COM 2019 -1]. The aims of this include combating climate change, ensuring environmental protection and preserving biodiversity. The knowledge about organic farming, despite being plentiful, clearly still needs to be further augmented to make practices yet more sustainable and yet more productive.
One of the main tasks in the search for environmentally friendly crop-growing methods is to increase soil fertility by improving its physical, chemical and biological parameters. One of the main components determining soil fertility is the organic matter (OM) content.
Sustainable management of organic matter in agriculture relies on increasing the contribution of MO to the soil while reducing its losses (Rober 2001).
Added reference:
- Brevik, E.C. The potential impact of climate change on soil properties and processes and corresponding influence on food security. Agriculture 2013, 3, 398–417.
- Law, B.E.; Hudiburg, T.W.; Berner, L.T.; Kent, J.J.; Buotte, P.C.; Harmon, M.E. Land use strategies to mitigate climate change in carbon dense temperate forests. Proc. Acad. Sci.U. S. A. 2018, 115, 3663–3668.
- Kopittke, P.M.; Menzies, N.W.; Wang, P.; McKenna, B.A.; Lombi, E. Soil and the inten[1]sification of agriculture for global food security. Int. 2019, 132, 105078.
- Hossain, A.; Krupnik, T.J.; Timsina, J.; Mahboob, M.G.; Chaki, A.K.; Farooq, M. Agricultural land degradation: processes and problems undermining future food security. In Environment, Climate, Plant and Vegetation Growt, Fahad, S., Hasanuzzaman, M., Alam, M., Ullah, H., Saeed, M., Ali Khan, I., Adnan, M. Eds.; Springer International Publishing: Cham, Switzerland, 2020; pp. 17–61.
- Scherr, S.J. Soil Degradation: A Threat to Developing-country Food Security by 2020? Food Policy Research Inst: Washington, USA, 1999.
- Hari, V.; Rakovec, O.; Markonis, Y.; Hanel, M.; Kumar, R. Increased future occurrences of the exceptional 2018–2019 central european drought under global warming. Rep. 2020, 10 (1), 12207.
- Ray, D.K.; West, P.C.; Clark, M.; Gerber, J.S.; Prishchepov, A.V.; Chatterjee, S. Climate change has likely already affected global food production. PLoS One 2019,14(5), e0217148.
- COM, 640 final. Communication from the Commission to the European Parliament, the Council, the European Economic and Social Committee and the Committee of the Regions. The European Green Deal, Brussels Available online: https://eur-lex.europa.eu/resource.html?uri=cellar:b828d165-1c22-11ea-8c1f-01aa75ed71a1.0016.02/DOC_1&format=PDF, (accessed on 11 December 2019).
- Robert, M. Soil Carbon Sequestration for Improved Land Management. In World Soil Resources Report; FAO: Rome, Italy, 2001; p. 75.
The aim was changed:
There was:
The aim of this study was to determine the effect that the long-term annual application of types of soil fertility agents (exogenous organic matter: 1. manure, 2. straw in combination with nitrogen fertilization and liming and 3. the addition of biostimulants) had on organic matter properties, including humic acid properties. Con-ducting such a wide spectrum of fertilization variants under identical soil and climatic conditions in a single experiment will also allow for the identification of how bio-fertilizers (UGmax, preparations increasingly used in fertilization) shape soil organic matter properties.
There is:
The aim of this study was to determine the effect of long-term use of: (1) exogenous organic matter (manure, straw), (2) mineral fertilization (CO (NH2) 2, (3) liming and (4) a biostimulator (UGmax) on the properties of organic matter including the properties of humic acids It was assumed that such a wide spectrum of fertilization variants, carried out in one experiment under the same soil and climatic conditions, would also allow to determine the role of individual fertilizers in shaping the properties of soil organic matter.
Materials and Methods
The Material and Methods section has been improved
The title subsection „Materials” was change to „Description of the study materials”
Description of the study materials was completed
There was:
The ten-year pot experiment was set up as single-factor pot experiment with four rep-lications. PVC pots (V=14.72 dm3; h=30 cm, r=12.5 cm) with perforated bottoms were filled with soil samples (15 kg each). The pots were placed in the field in a completely random manner and dug-in to a depth of 25 cm. …
Each soil application was performed by mixing the substance (except for mulch) into the soil. Until the next application, no activities were performed in the pots except for removing emerging vegetation using glyphosate. The pots were exposed directly to the weather.
There is:
The soil sample was taken from the tilled layer of an arable field (0–20 cm) where winter wheat was grown in monoculture. According to the WRB classification [IUSS], the sampled soil was classified as Luvisol, with a granulometric composition characteristic for light clay. The ten-year pot experiment was set up as single-factor pot experiment with four replications. PVC pots (V=14.72 dm3; h=30 cm, r=12.5 cm) with perforated bottoms were filled with soil samples (15 kg each). The pots were placed in the field in a completely random manner and dug-in to a depth of 25 cm. The experiment was located at Kicko ( N:52036’30,1’’ and E:18024’00,2’’) in the Kuyavian-Pomeranian Voivodeship, Poland. Soil fertility agents were applied in the following amounts in the first decade of September each year:
Each soil application was performed by mixing the substance (except for mulch) into the soil. Until the next application, no activities were performed in the pots except for removing emerging vegetation using glyphosate and keeping the surface free from vegetation. The pots were exposed directly to the weather. The test samples were collected once after 10 years of the experiment, by the vases liquidation. Soil samples dried in room temperature, and sieved (2 mm).
Added references:
IUSS Working Group WRB. World Reference Base for Soil Resources 2014, Update 2015. International for Soil Classification System for Naming Soil and Creating Legends for Soil Maps; World Soil Resources Reports No 106; FAO: Rome, Italy, 2015.
Methods:
Due to the content of the other reviews and the requirements of the editorial office, the methodological description was not shortened, while references were added to each method. Due to the above, the literature list has changed by the following items:
Schnitzer, M.; Khan, S.U. Humic substances in the environment; Marcel Dekker: New York, USA, 1972.
Debska, B.; Spychaj-Fabisiak, E.; Szulc,W.; Gaj, R.; Banach-Szott, M. EPR Spectroscopy as a tool to characterize the maturity degree of humic acids. Materials 2021, 14, 3410. https://doi.org/10.3390/ma14123410
Ward, J.H. Hierarchical grouping to optimize an objective function. Journal of the American Statistical Association, 1963, 58, 236–244.
Gomez, K.A.; Gomez, A.A. Statistical Procedures for Agricultural Research; John Wiley and Sons: New York, USA, 1983.
The abbreviation Nmin is explained and next to UGmax, a literature entry with the description of the composition of UGmax has been added.
Results and discussion
As suggested by the Reviewer, In order to increase the readability and correct description of the results, the Annova statistical analysis was additionally carried out (Tables 2, 4 - 8). Section Results and disscusion was corrected according to the statistical results.
As this chapter contains a description of the results and a discussion, the authors decided that the so-called a foreword to each of the discussed parameters will make it easier for the reader to understand the obtained dependencies.
Reduction of organic carbon in the soil samples mixed only with calcium oxide was explain:
There was: The results confirm the finding of Aye et al. [41] that long-term liming (34 years) reduces TOC content.
There is: The results confirm the finding of Aye et al. [41] that long-term liming (34 years) reduces TOC content. Liming enhances the OC mineralization processes due to the increase of pH in the soil.
Table 3
The units presented in the Table 3 relate to the parameters that were subject to correlation analysis. Parameters of the fractional composition of organic matter expressed in mg / kg and as a percentage share were correlated. Hence the presence of units is essential.
Missing figures (1-5) have been added to the text.
Conclusions
It was added in the conclusions that the obtained results relate to the pot experiment.
Reviewer 2 Report
Review Comments for Agriculture manuscript agriculture-1893873, entitled "Improving soil fertility and carbon sequestration through the application of exogenous organic matter and biostimulants...", submitted by Debska et al. (2022).
General Comments
This manuscript reports on the application of exogenous organic matter (i.e., manure, straw, and biostimulants) improved soil fertility and carbon sequestration. This manuscript is a good scientific contribution. However, in its present form, this manuscript has numerous limitations, including, but not limited to, the following: i) overall writing quality that needs improvement, ii) unclear description of statistical analyses performed, and iii) many claims that are not supported by the presented results. With this, I would recommend "Reject and Resubmit".
Specific Comments
Title
-Because the biostimulants were included in exogenous organic matter, I suggested to delete it.
Abstract
-Moveing this sentence to the ending of this paragraph.
-Positive correlations? Negative correlations? p values? R or R-square values?
-What is the A1?
Introduction
-Adding references in those sentences.
-Lacking right parentheses.
-Deleting left parentheses.
-Could you provide some clear hypotheses on your study?
Materials and Methods
-Adding line under this row in Table 1.
-Please use capital letter here, i.e., 'Mulch', 'Chopped', and 'Manare' in Table 1.
-Is this a Table? Deleting the dot between 't' and 'ha-1' as well as 'g' and 'pot-1'.
-Adding reference regarding those measurment methods.
-Why did not you perform ANOVA and multiple comparisons on target variables among different treatment in Tables 2 and 4-8?
Results and Discussion
-My main concern on this section is that the results have not been analysed by the ANOVA analysis and multiple comparisons, so that many claims that are not supported statistically. With this, I think that this section needs to be completely rewritten following the results of the statistical analyses conducted.
-Why can you obtain this result? Indeed, you have not perform ANOVA analysis and multiple comparisons.
-Statistically the highest? p < 0.05?
-Please delete the line in this and following (Tables 2-9) tables and strictly implement the three-line table for scientific research. Additionally, I suggested you to make conversions of those tables to figures in order to improve the readability.
-Table 3: Is this R or R-square values? The correlations are significant? (p < 0.05? p < 0.01? p < 0.001?).
-Where is Figure1-5? Sorry, I can not find them.
-Table 9: Is this R or R-square values? The correlations are significant? (p < 0.05? p < 0.01? p < 0.001?).
Conclusions
-This section will need to be completely revised once the results are revised to properly report data/means following the results of the statistical analyses conducted.
P.S. The Specific Comments have been included on an annotated manuscript that was submitted with these review comments in the submission system. Please check it out.

Author Response
Dear Reviewer,
The paper's Authors wish to thank for all the precious comments and guidelines.
I hope that the corrections introduced are satisfactory. All the changes have been marked in the text.
With best regards,
Magdalena Banach-Szott
The title of the work has been redrafted as suggested by another Reviewer. Due to the assumptions and purpose of the work, the word biostimulants has not been removed from the title
Abstract
Sentece” Conducting such a wide spectrum of fertilization variants under identical soil and climatic conditions in a single experiment will also allow for the identification of how bio-fertilizers (UGmax, preparations increasingly used in fertilization) shape soil organic matter properties” has been removed from the abstract.
There was: The content of dissolved organic carbon (DOC) ranged from 124.6 to 286.1 mg kg˗1 and correlated strongly with TOC content.
There is: The content of dissolved organic carbon (DOC) ranged from 124.6 to 286.1 mg kg˗1 and correlated strongly positively with TOC content.
p values are given in the description of the statistical methods and added to the tables.
A1 is explained under Table 1. Above abbreviation has been removed from the abstract
All the changes made are marked in the text).
Introduction
In the introduction as suggested by the Reviewer and the missing literatures was supplemented.
Materials and Methods
Used capital letter here, i.e., 'Mulch', 'Chopped', and 'Manare' in Table 1.
References were added to each method. Due to the above, the literature list has changed by the following items:
Schnitzer, M.; Khan, S.U. Humic substances in the environment; Marcel Dekker: New York, USA, 1972.
Debska, B.; Spychaj-Fabisiak, E.; Szulc,W.; Gaj, R.; Banach-Szott, M. EPR Spectroscopy as a tool to characterize the maturity degree of humic acids. Materials 2021, 14, 3410. https://doi.org/10.3390/ma14123410
Ward, J.H. Hierarchical grouping to optimize an objective function. Journal of the American Statistical Association, 1963, 58, 236–244.
Gomez, K.A.; Gomez, A.A. Statistical Procedures for Agricultural Research; John Wiley and Sons: New York, USA, 1983.
Annova statistical analysis was performed (Tables 2, 4-8), the statistical description was supplemented in the Material and methods section:
„To determine the significance of differences of the parameters, the analysis of variance (ANOVA) for p<0.05. The significance of the effect of the factors and interactions was verified with test F, and the significance of differences between the values of respective traits with the post–hoc Tukey test at p = 0.05.” Statistical calculations were performed in three repetitions.
Results and discussion
As suggested by the Reviewer, In order to increase the readability and correct description of the results, the Annova statistical analysis was additionally carried out (Tables 2, 4 - 8). Section Results and discussion was corrected according to the statistical results.
The presented conclusions did not require correction as they coincided with the results of the statistical analysis.
Missing figures (1-5) have been added to the text.
Stylistic and editorial errors have been corrected.
All the changes made are marked in the text.
Reviewer 3 Report
Please open the attached file which contains the comments and inquiries.

Author Response
Dear Reviewer,
The paper's Authors wish to thank for all the precious comments and guidelines.
I hope that the corrections introduced are satisfactory. All the changes have been marked in the text.
With best regards,
Magdalena Banach-Szott
The ten-year pot experiment was set up as single-factor pot experiment with four replications. After 10 years of the experiment, samples were taken from each replicate for basic analyzes (removal of vases). Three repetitions were used for the statistical calculations
As suggested by the Reviewer, an analysis of variance (ANOVA) was performed to determine the significance of parameter differences (Tables 2, 4-8).
All the changes made are marked in the text.
Reviewer 4 Report
The manuscript is not well constructed and lacks seriousness during the submission. Find the comments below, mostly technical.
1. The 'Introduction' part is not well written. It should reflect a story of why the study is important and considered here, and not just giving some previous references. This must be improved.
2. What is the full form of CHAs/CFAs ? Give the full form in the introduction.
3. Table 1 is not needed. It can be written in the text clearly.
4. If the study is a 10-year experiment, as mentioned in the method, then what is the sampling frequency in a single year ?
5. No Cluster analysis and Dendrogram result is given.
6. The article mentions about 3 figures, but no Figure file is given.
7. Number of tables is way too much and must be converted into some graphical representations. At least 2-3 tables can be converted to a graphical scheme.
8. The discussion is poor and needs revision. At present, it looks like detailed and elaborated results.
Author Response
Dear Reviewer,
The paper's Authors wish to thank for all the precious comments and guidelines.
I hope that the corrections introduced are satisfactory. All the changes have been marked in the text.
With best regards,
Magdalena Banach-Szott
Introduction
In the introduction as suggested by the Reviewer and the missing literatures was supplemented.
All the changes made are marked in the text.
The full form of CHAs/CFAs is already in the Introduction section: “An equally important parameter used to determine the quality of organic matter is CHAs/CFAs – the ratio of carbon content in humic acids to carbon content in fulvic acids [10]”
Materials and Methods – has been supplemented.
There was:
Each soil application was performed by mixing the substance (except for mulch) into the soil. Until the next application, no activities were performed in the pots except for removing emerging vegetation using glyphosate. The pots were exposed directly to the weather.
There is: Each soil application was performed by mixing the substance (except for mulch) into the soil. Until the next application, no activities were performed in the pots except for removing emerging vegetation using glyphosate and keeping the surface free from vegetation. The pots were exposed directly to the weather. The test samples were collected once after 10 years of the experiment, by the vases liquidation. Soil samples dried in room temperature, and sieved (2 mm).
Results and discussion
In order to increase the readability and correct description of the results, the Annova statistical analysis was additionally carried out (the statistical description was supplemented in the Material and methods section, results presented in Tables 2, 4 - 8). Section Results and disscusion was corrected according to the statistical results.
Missing figures (1-5) have been added to the text.
Stylistic and editorial errors have been corrected.
Round 2
Reviewer 1 Report
Thanks to the authors as they addressed all the comments.
Author Response
Dear Reviewer,
The Authors would like to thank you for all comments and guidelines and for the possibility of publishing the article in Agriculture.
With best regards,
Magdalena Banach-Szott
Reviewer 2 Report
Dear Editor and Authors:
After careful revision by the authors of the manuscript, I recommend that the article can be published in Agriculture. The editing of the manuscript in relation to my queries is fully satisfactory.
Best wish.
Author Response

(The authors gave the same response as above.)

Reviewer 4 Report
Hello,
It seems that the raised issues have been addressed and improved from the previous version of the manuscript, yet it needs some minor addition to the result and discussion section. Please discuss the results obtained from FTIR and dendrogram. At present these are result statements. With these additions, I think the manuscript can be processed further.
Author Response
Dear Reviewer,
The paper's Authors wish to thank for all the precious comments and guidelines.
I hope that the corrections introduced are satisfactory. All the changes have been marked in the text.
With best regards,
Magdalena Banach-Szott
Results and discussion
FT-IR spectra of humic acids
There was:
The figure 3 shows examples of the infrared (IR) spectra of humic acids (of soil without additives and of soil incubated with straw left on its surface). The FT-IR spectra of the analyzed humic acids were characterized by the presence of the following absorption bands: 3400–3100 cm˗1, corresponding to O–H stretching of alcohols, phenols and acids, N–H stretching; 3100–3000 cm˗1, associated with the presence of C–H groups of aromatic and alicyclic compounds; 2960–2920 and 2850 cm˗1, corresponding to the asymmetric and symmetric C–H stretching of the CH3 and CH2 group (the intensity of these bands is taken as an indicator of the aliphaticity of HAs); 1730–1710 cm˗1 band, indicating the presence of C=O stretching of carboxyl, aldehyde, ketone group; 1660–1620 cm˗1 C=O stretching of amide groups; 1610–1600 cm˗1 indicates the presence of C–C stretching of aromatic rings; 1550–1530 cm˗1 N–H deformation, C=N stretching (amide II bands); 1520–1500 cm˗1 C–C stretching of aromatic rings; 1460–1440 cm˗1 corresponds to C–H bonds – asymmetric of CH3 and CH2; 1420–1400 cm˗1 C–O stretching and OH deformation of phenol groups; 1380–1320 cm˗1 C–N aromatic amine, COO, C–H stretching; 1280–1200 cm˗1 corresponds to C–O bonds stretching of aryl ethers, esters and phenols and 1160–1030 cm˗1 C–O stretching alcohols, ethers and polysaccharides [32,36,40]. It should be emphasized that, regardless of variant, the band in the 1730–1710 cm˗1 range, which is associated with the presence of C=O, was particularly intense; this indicates a high degree of oxidation of the tested humic acids (which is consistent with, among other things, the results of the elemental composition – O/C and ). Moreover, this band exhibited the greatest differences in intensity. The intensity of the 1730–1710 cm˗1 band increased in the following order:
A1 < A2 = B = G < C = F = K < H = E =D
The course of the FT-IR spectra also exhibited slight changes in the intensity of bands in the 2960–2920 and 2850 cm˗1 ranges (the bands assumed as the aliphaticity index) and the 1160–1030 cm˗1 bands.
There is:
The figure 3 shows examples of the infrared (IR) spectra of humic acids (of soil without additives and of soil incubated with straw left on its surface). The FT-IR spectra of the analyzed humic acids were characterized by the presence of the following absorption bands: 3400–3100 cm˗1, corresponding to O–H stretching of alcohols, phenols and acids, N–H stretching; 3100–3000 cm˗1, associated with the presence of C–H groups of aromatic and alicyclic compounds; 2960–2920 and 2850 cm˗1, corresponding to the asymmetric and symmetric C–H stretching of the CH3 and CH2 group (the intensity of these bands is taken as an indicator of the aliphaticity of HAs); 1730–1710 cm˗1 band, indicating the presence of C=O stretching of carboxyl, aldehyde, ketone group; 1660–1620 cm˗1 C=O stretching of amide groups; 1610–1600 cm˗1 indicates the presence of C–C stretching of aromatic rings; 1550–1530 cm˗1 N–H deformation, C=N stretching (amide II bands); 1520–1500 cm˗1 C–C stretching of aromatic rings; 1460–1440 cm˗1 corresponds to C–H bonds – asymmetric of CH3 and CH2; 1420–1400 cm˗1 C–O stretching and OH deformation of phenol groups; 1380–1320 cm˗1 C–N aromatic amine, COO, C–H stretching; 1280–1200 cm˗1 corresponds to C–O bonds stretching of aryl ethers, esters and phenols and 1160–1030 cm˗1 C–O stretching alcohols, ethers and polysaccharides [32,36,40]. It should be emphasized that, regardless of variant, the band in the 1730–1710 cm˗1 range, which is associated with the presence of C=O, was particularly intense; this indicates a high degree of oxidation of the tested humic acids (which is consistent with, among other things, the results of the elemental composition – O/C and ). Moreover, this band exhibited the greatest differences in intensity. The intensity of the 1730–1710 cm˗1 band increased in the following order:
A1 < A2 = B = G < C = F = K < H = E =D
The course of the FT-IR spectra also exhibited slight changes in the intensity of bands in the 2960–2920 and 2850 cm˗1 ranges (the bands assumed as the aliphaticity index) and the 1160–1030 cm˗1 bands. The lowest intensity of the above-mentioned bands were characterized the HAs in the soil with the addition of UGmax (variants D and E), HAs with the lowest values of the H/C ratio.
As in the literature reports [17,32,40] humic acids with higher band intensity in the range of 1730 - 1710 cm-1 and smaller bands in the range of 2960–2920 and 2850 cm˗1 and in the range of 1160–1030 cm˗1, are characterized by a higher "degree of maturity". Thus, the FT-IR spectra indicate a higher degree of maturity of humic acids in the soil with the addition of UGmax in comparison to the HAs of the other variants.
Cluster analysis
There was:
The parameters correlated with the PCA1 component were used in the cluster analysis. Cluster analysis distinguished two main groups with similar properties (Fig. 5):
- Group I comprises the HAs isolated from the soil samples with UGmax,
- Group II comprises the soil HAs of the remaining variants. However, this group includes subgroups that comprise:
- Subgroup I HAs of the soil without additives, the soil mixed with CaO and the soil mixed with chopped straw (variants B and F);
- Subgroup II HAs of the soil mixed with manure, the soil from the 10–20-cm layer onto which a mulch of chopped straw was applied, and the soil with chopped straw and CaO.
The HAs of the surface layer covered with straw mulch lying outside the above-mentioned subgroups.
There is:
The parameters correlated with the PCA1 component were used in the cluster analysis. To obtain complete information on the differences (similarities) of humic acids depending on the kind of fertilizer, the cluster analysis was applied. Humic acids with similar properties are located on dendrograms in homogenous groups.
Cluster analysis distinguished two main groups with similar properties (Fig. 5):
- Group I comprises the HAs isolated from the soil samples with UGmax,
- Group II comprises the soil HAs of the remaining variants. However, this group includes subgroups that comprise:
- Subgroup I HAs of the soil without additives (K), the soil mixed with CaO (H) and the soil mixed with chopped straw (variants C and F);
- Subgroup II HAs of the soil mixed with manure, the soil from the 10–20-cm layer onto which a mulch of chopped straw was applied (A2), and the soil with chopped straw and CaO (B) and variant G (soil mixed with manure).
The HAs of the surface layer covered with straw mulch lying outside the above-mentioned subgroups (variant A1). The obtained results indicate a significant role in tayloring the HAs properties of the UGmax biostimulator. According to the literature review [13,43,48], the introduction of biostimulants into the soil increases significantly the enzymatic activity of soils. Enzymatic activity is one of the important factors determining the intensification of humification processes. Increase in the intensity of the organic matter humification process causes an increase in the "maturity" of HAs.